# Multi-source Domain Adaptation Techniques for Mitigating Batch Effects: A Comparative Study

**Rohan Panda**[1]                                                    F20170487@HYDERABAD.BITS-PILANI.AC.IN
**Sunil Vasu Kalmady**[2,3]                                                      KALMADY@UALBERTA.CA
**Russ Greiner**[3,4]                                                             RGREINER@UALBERTA.CA

[1] *Electrical & Electronics Engineering, Birla Institute of Technology and Science, Pilani, India*
[2] *Canadian VIGOUR Centre, University of Alberta, Edmonton, Canada*
[3] *Department of Computing Science, University of Alberta, Edmonton, Canada*
[4] *Alberta Machine Intelligence Institute, Edmonton, Alberta, Canada*

**Editors:** Under Review for MIDL 2021

## Abstract

The past decade has seen an increasing number of applications of deep learning (DL) techniques to biomedical fields, especially in neuroimaging-based analysis. Such DL-based methods are generally data-intensive and require large number of training instances, which might be infeasible to acquire from a single acquisition site, especially for data such as fMRI scans, due to the time and costs that they demand. We can attempt to address this issue by combining fMRI data from various sites, thereby creating a bigger heterogeneous dataset. Unfortunately, the inherent differences in the combined data, known as batch effects, often hampers learning a model. To mitigate this issue, techniques such as multi-source domain adaptation (MSDA) aim at learning an effective classification function that uses (learned) domain-invariant latent features. This paper analyzes and compares the performance of various popular MSDA methods (MDAN, DARN, MDMN, $M^3SDA$) at predicting different labels (illness, age and sex) of images from several public rs-fMRI datasets: ABIDE I and ADHD-200. It also evaluates the impact of various conditions such as: class imbalance, number of sites along with a comparison of the degree of adaptation of each of the methods, thereby presenting the effectiveness of MSDA models in neuroimaging-based applications.
**Keywords:** Resting-state fMRI, multi-source domain adaptation, batch effects, deep learning, ADHD, ASD

## 1. Introduction

### 1.1. Motivation and Background

With recent developments in brain imaging technology, data in the form of functional Magnetic Resonance Imaging (fMRI), electroencephalography (EEG), and Magnetoencephalography (MEG) have become widely available, which can be helpful in conducting various diagnostic and predictive analyses. Owing to the spatio-temporal nature of fMRI data, which allows for extensive information extraction, there has been a steady rise in the applications of various deep learning (DL) strategies applied to fMRI data to classify or predict mental illnesses (*e.g.*, Alzheimer's, ADHD, Schizophrenia, etc.), brain states (*e.g.*, sleep stages, task-based activity, etc.) or patient demographics (*e.g.*, age, gender, IQ, etc.).

DL models are data-intensive in nature and tend to work better as we increase the size of the data available for training. However, owing to the difficulties related to the acquisition of fMRI data, building a large dataset is often infeasible, expensive and time-consuming.

A general workaround involves building a large dataset by combining data from various acquisition sites for a particular research task. This, however, leads to another problem that arises as the data was collected from multiple sites, which means this can involve different acquisition methods, equipment, demographic of patients, methodology, etc. The models can be trained on a dataset that simply contains all of these instances, without any modifications. However, this method ignores these differences, that might hamper the model generalizability. The basic reason for such variations is the differences in the probability distributions of data and label across sites, which are generally termed as domain shift, and also, batch effects (Dundar et al., 2007).

Recent works in various domains have focused on developing methods to mitigate such issues, including domain adaptation (DA) techniques, which aim at building a generalized model that can learn from the multiple given source sites to produce a model that can perform reasonably well on a new, yet related, target site. The existing DA techniques have varied approaches based on factors like: number of source sites (single-source DA, multi-source DA), labels availability in target domain (unsupervised, semi-supervised, supervised DA), method of domain adaptation (discrepancy, adversarial, reconstruction based), etc. Which technique performs best can depend on the objective at hand, and the type of datasets that have been used. This motivates our work, here, to study the existing DA methods and their performance when dealing with multi-site biomedical (in this case, resting-state fMRI) data.

### 1.2. Related Works

In the past decade there have been various new techniques of that apply DL tools to fMRI data, to develop predictive models based on numerous objectives.

Many systems view raw fMRI data as a sequence of 3-dimensional data, motivating various techniques which use 3D convolutions to build models such as using 3D-CNN to predict ADHD using fMRI and structural MRI (Zou et al., 2017), extracting features using 3D-Convolutional Autoencoders for mild Traumatic Brain Injury recognition (Zhao et al., 2017), predicting Schizophrenia using 3D-CNN, development of 2-channel 3D DNN for autism spectrum disorder (ASD) classification (Li et al., 2018b). Though such methods allow for maximal information extraction, the deep models are computationally very expensive and generally infeasible. To mitigate this issue, functional connectivity matrices (Lynall et al., 2010) are popularly used and are found to be a good replacement, making the training computationally feasible and also providing a way to interpret the results. Some noteworthy results using FCMs include classification of Schizophrenia patients using various DL methods (Arbabshirani et al., 2013; Shen et al., 2010; Yan et al., 2017), prediction of other illnesses such as attention deficit hyperactivity disorder (ADHD) (Riaz et al., 2020), Alzheimer's (Ju et al., 2017), ASD (Li et al., 2018a; Saeed et al., 2019) and Mild Cognitive Impairment (Chen et al., 2016). There have also been classifications of other brain states, such as suicidal behavior (Gosnell et al., 2019), chronic pain (Santana et al.,

2019), migraine (Chong et al., 2017) and demographics such as age (Pruett Jr et al., 2015) or gender (Fan et al., 2020), etc.

While the works mentioned above have shown impressive results, none addressed the issue of batch effects. However, there have been few recent methods that have tried to deal with batch effects in different ways. Olivetti et al. (2012) was one of the first to investigate batch effects in rs-fMRI datasets (ADHD-200) using extremely randomized trees along with dissimilarity representation. Vega and Greiner (2018) studied the impact of classical techniques such as covariate, z-score normalization, and whitening on batch effects. Wang et al. (2019) explored ways to use low-Rank domain adaptation to reduce existing biases on multi-site fMRI dataset. Recent approaches include, transport-based joint distribution alignment (Zhang et al., 2020), federated learning (Li et al., 2020) and conditional autoencoder (Fader Networks) (Pominova et al., 2020).

It is therefore useful to have a comparative survey of the performances of various existing MSDA techniques applied to solve the batch effects in multi-site fMRI datasets, to understand the benefits and limitations of DA approaches.

## 2. Domain Adaptation Methods

We define the common objective of MSDA techniques as follows: Given a collection of labelled source-domain data $\mathfrak{D}_s = \{(x_s^i, y_s^i)\}_{i=1}^{N_s}$ $\forall s \in \{1, \ldots, S\}$ and a collection of unlabelled target-domain data $\mathfrak{D}_T = \{x_T^i\}_{i=1}^{N_T}$ (where $x_s^i$, $x_T^i \in X$ and $y_s^i \in Y$) the goal is to build a classifier that can use information from the source domains to help produce models that can perform accurate classifications in the target domain (Zhao et al., 2020). For our experiments, We take one of $S$ domains as the target domain (by convention, this is the domain indexed by $S$) and use the others as source domains ($s \in \{1, ..., S-1\}$). Generally, MSDA techniques employ different strategies of transforming the target domain distribution into the source domain distributions to tackle the issue of batch effects. We allow the marginal probability distributions $P_X$ to be different across domains, but require the conditional probability distributions $P_{Y|X}$ remain the same. Below is a short introduction to the various methods used in our experiments.

**Domain Adversarial Neural Networks (DANN)**: Considered as one of the fundamental models in DA, DANN(Ajakan et al., 2014) is a single-source DA technique – the only single-source DA method included in the comparison. DANN's architecture is similar to Figure 1(b), except that all the sources' data are combined as used as a single big source.

**Multi-source Domain Adversarial Networks (MDAN)**: MDAN can be seen as a natural extension of DANN for MSDA problems. Its feature extractor and label classifier are essentially the same as DANN's, but MDAN uses one domain adapter $M_{d_i}$ for each of the $S-1$ source domains. Zhao et al. (2018) introduces two versions of MDAN: hard-max and soft-max variants. We use just the soft-max variant as it is shown to provide better generalization in (Zhao et al., 2018).

**Domain AggRegation Networks (DARN)**: One of MSDA's main challenges is that it needs to include source sites based on the target site, in a way that minimizes the negative

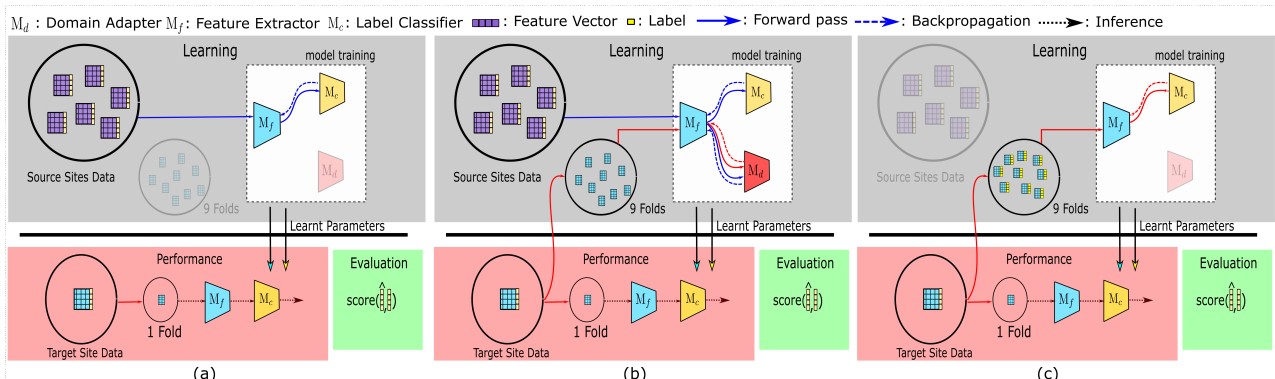

Figure 1: The training & evaluation pipelines used in this work. (a) represents the SRC model, used as the baseline. All other MSDA models can be generalised to be using (b). Finally, to compare the accuracy achieved with target-only data – the TAR model – is shown in (c).

transfer while preserving as many training instances as possible. To tackle this issue, DARN dynamically selects source sites and gives the sites varied importance based on their label classification losses. This is possible by solving the Lagrangian dual of the objective that needs to be optimized by utilizing binary search strategies.

**Multi-Domain Matching Networks (MDMN)**: MDMN tackles MSDA by first projecting features into a shared feature space. By computing, then using a degree of similarity between the target and source sites, MDMN merges similar sites together to construct the shared feature space, while reducing the negative transfer by keeping dissimilar sites distant. This model tackles this objective by using a loss function based on Wasserstein distance and a special training paradigm as described in (Li et al., 2018c).

**Moment Matching for MSDA($M^3$SDA)**: Unlike the previous discussed models, $M^3$SDA aligns target domain with source domains while simultaneously aligning source sites amongst themselves. Furthermore, it tackles this issue by utilizing the feature distribution moments instead of the raw input features for adaptation, which provides a certain robustness and a statistical advantage in MSDA. Peng et al. (2019) introduces an extension of $M^3$SDA, called $M^3$SDA-$\beta$, which they demonstrate performs better against overfitting and provides better generalization. We therefore use M3SDA-$\beta$ to understand the model's performance on neuroimaging data.

Appendix A provides more information about each of these architectures.

## 3. Methodology

### 3.1. Datasets and Tasks

This study uses two different publicly available datasets for training and evaluation, selected on the basis of the number of total scans available, the number of sites of data acquisition, and their frequent usage in the research community.

The first dataset consists of rs-fMRI scans from the ABIDE I dataset (Craddock et al., 2013a), including 530 control instances (tagged as typical controls, TC) and 505 instances collected from subjects suffering with Autism Spectrum Disorder (ASD), which have been acquired from 17 different sites. Appendix B shows the phenotypic information and pre-processing steps used in the dataset.

The ADHD-200 dataset is our second multi-site fMRI dataset, which has been compiled from 8 different sites, and contains 1516 rs-fMRI scans in total: 842 scans from control subjects, and 674 from subjects who suffer from Attention Deficit Hyperactivity Disorder (ADHD). Again, see Appendix B for further information.

For each dataset, we run three different classification tasks, using three different labels: (1) the respective mental illnesses, between illness and control samples; and binary classification of two phenotypic labels, (2) sex and (3) age (old versus young, w.r.t. the global median calculated separately for each of the two datasets).

### 3.2. Functional Connectivity Matrix and Feature Extraction

Functional connectivity is defined as the temporal dependency of spatially-remote neurophysiological events (Van Den Heuvel and Pol, 2010). It computes the level of co-activation between two spatially separate regions of interest (ROIs) in the brain, based on the mean time-series extracted these ROIs. Each ROI is pre-defined using some atlas or template. Here, we use the Automatic Anatomical Labelling (AAL) atlas (Tzourio-Mazoyer et al., 2002), which partitions the brain into $^{AR4:}$ ~~119~~116 different non-overlapping ROIs.

We then calculate the functional connectivity matrix (FCM) using Pearson's correlation coefficient between each pair of time-series which results in a $^{AR4:}$ ~~119 × 119~~ $116 \times 116$ matrix. Since the diagonal of this matrix is redundant and the matrix is symmetric, the diagonal is dropped and the upper triangle of the matrix is flattened to finally produce a vector of size $^{AR4:}$ ~~$\binom{119}{2}$~~ $\binom{116}{2}$ for each rs-fMRI scan, which is used as the input data to various models in this study.

### 3.3. Training and Testing Settings

The MSDA models require labeled data flowing in from multiple source domains, as well as a batch of unlabeled data from the target domain. To accommodate this, we first take a single source is the target domain, and consider the remaining sites as different source domains. The target domain is then split using a $^{AR2:}$ stratified 10-fold cross-validation strategy, wherein a single fold is kept aside for testing while the remaining 9 folds are used (without their labels) to provide the unlabeled target domain data required for the unsupervised-MSDA methods. All datapoints from the source sites are fed into the model along with their labels during training. We repeat the training and testing for each fold and for each site, then report the average accuracies as the results. Figure 1 shows this pipeline.

To compare the performance of MSDA models (Figure 1(b)), the SRC model (see Figure 1(a)) is used as the baseline model. In this setting, data from all the source sites are combined and are treated as one big dataset; that is, no target site data is used. Also, the TAR model uses only the (labeled) target site data (and no source sites), in a $^{AR2:}$ stratified 10-fold CV setting to maintain the class distribution in all the folds. This model shows the baseline performance when only the target site information is available.

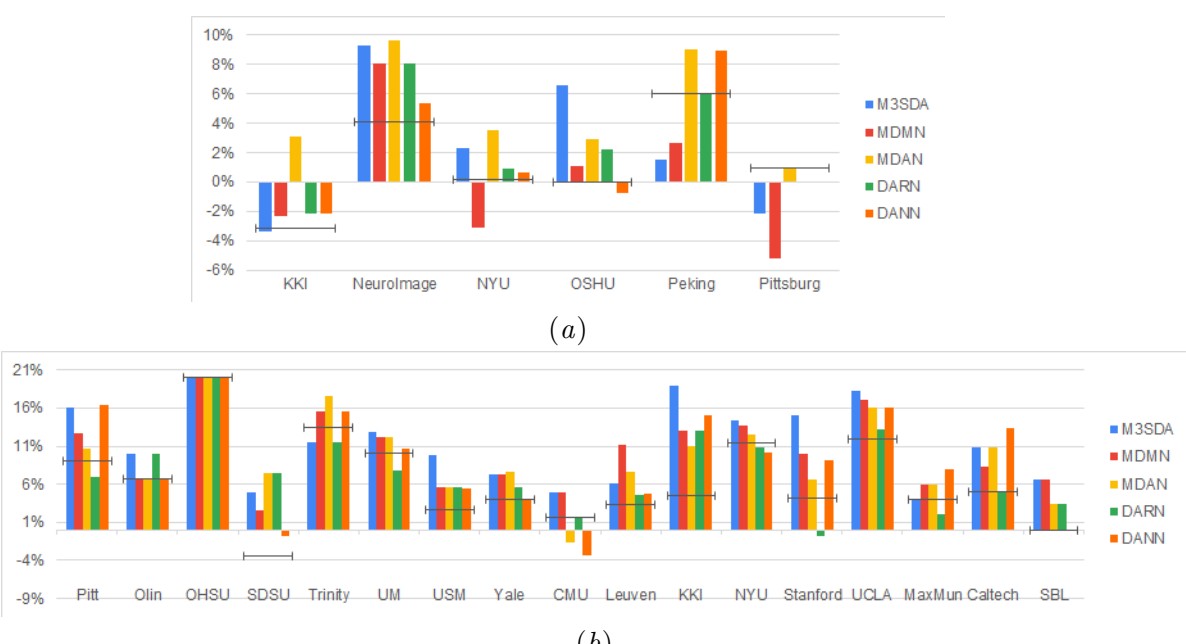

($a$)

($b$)

Figure 2: Site-wise performance of MSDA and baseline models in $^{AR2,\ AR4:}$illness classification for $^{All:}$ ADHD-200 (a) and ABIDE I (b) . The height of each bar is the average of 10-Fold CV accuracies achieved when the particular site was chosen as the target site. To show how much better an MSDA model performs w.r.t. using only the target site data, and also w.r.t. using all source data without any domain adaptation, we then subtract the TAR model accuracy from every model's original accuracy, and $^{AR3:}$show the SRC model's scores as a black target-line.

## 4. Results

**Mental Illness**: Figure 2 and Figure 3 show that MSDA models produce classifiers that are more accurate than SRC and TAR. In case of ABIDE I, we find a significant increase in the accuracy scores from baseline (SRC) to the MSDA models, with the highest increase being in M$^3$SDA ( 5%). In the case of ADHD-200 data, the data was balanced (see Appendix D) and used for experimentation. In comparison to ABIDE, MSDA is only slightly more accurate than SRC ( 1-4%). MDAN (76.21%) has the highest increase in comparison to baseline (72.72%), while MDMN (71.54%) does not outperform baseline accuracy. We noted the positive impact of balancing the data (Appendix D) using oversampling for this dataset. This observation suggests that current MSDA models might be impacted by the class balance present in data used.

Figure 2 provides a deeper look at the site-wise performance of the models. For ABIDE I, Figure 2($b$)subfigure shows that generally the models performed better than TAR and SRC, with only a few exceptions. For site OHSU, no model was able to perform better than SRC, while in some sites such as SDSU, CMU and Stanford, a few models performed worse than TAR. For ADHD-200, Figure 2($a$)subfigure shows that sites such as KKI and Pittsburg have MSDA scores lower than TAR, however, in both of these sites, MDAN is able to perform better in comparison to TAR. M$^3$SDA has a higher accuracy in most (around 10 out

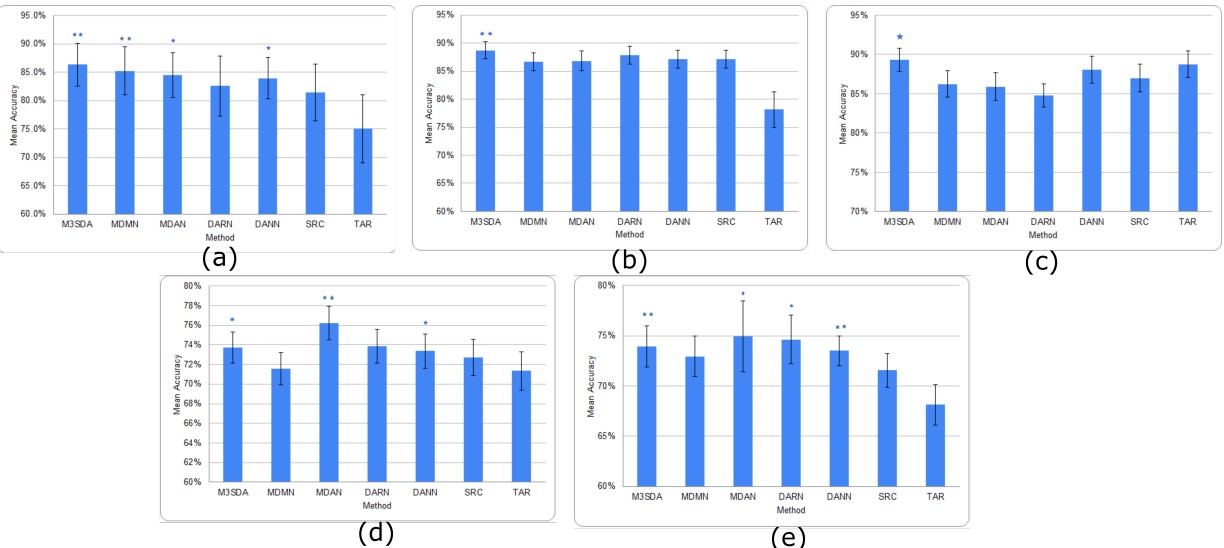

Figure 3: Average accuracies of all models across all sites. (a),(b),(c) depict the accuracies for illness, age and sex for ABIDE I respectively, while (d) and (e) are for illness and sex for ADHD-200 dataset. *(p<0.05) and **(p<0.01) depict the statistical significance between the MSDA methods and SRC models, each calculated using a paired-t test.

of 17) of the sites in ABIDE I, while in ADHD-200, MDAN seems to perform better with some consistency. Furthermore, the increase in accuracy using MSDA models differ from ABIDE I (16-20%) to ADHD-200 (8-10%).

**Age**: Figure 3(b) shows that applying DA models does not increase the classification performance over the baseline results. The baseline and MSDA models both have an accuracy of around $^{AR3:}$~~90-91~~86-88%. To explore whether this was due to class imbalance, we applied a strategy similar to the one in Section D, however that did not improve the results. It could be that demographic information such as age might be domain invariant, which would explain why MSDA models did not improve baseline performance.

**Sex**: Figure 3(c) presents the accuracy scores of the experiments on the ABIDE I data. While only M³SDA showed significant increase for ABIDE I, in ADHD-200 all the DA models except MDMN scored significantly better than baseline scores. While MDAN performs the best in ADHD-200, it is not so accurate when it comes to ABIDE I; moreover, we see that M³SDA is consistently accurate in both types of datasets.

The previous results show that the MSDA models perform better than simply combining all the source data and utilizing it without any adaptation. To explore how well the models harmonize the source sites, we ran experiments on MSDA models' ability to make features site-invariant. Figure 4 reports the results of a two-layered fully-connected network that was trained and tested to classify the sites based on input latent features in a 10-fold CV setting. We found that, in ABIDE I, the generalization of sites seems to be better than in the case of ADHD-200. $^{AR3:}$We observe that, though MSDA methods have lower accuracy in

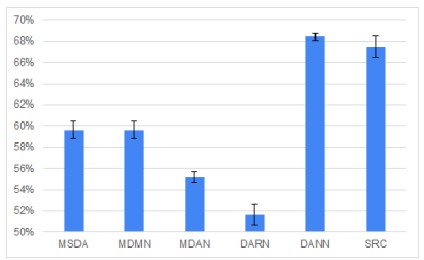
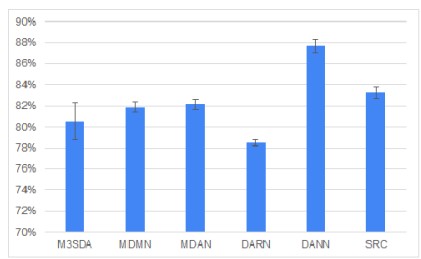

$(a)$ $\qquad\qquad\qquad\qquad\qquad\qquad$ $(b)$

Figure 4: Average accuracy values 10 fold CV of site classification using latent features learnt in various models for ABIDE I(a) and ADHD-200(b).

site classification, it is still greater than chance ($\frac{1}{17}$ for ABIDE Iand $\frac{1}{8}$ for ADHD-200). This is owing to the trade-off between harmonizing sites and retaining discriminatory information for the classification that each model must tackle. Since each model tries to achieve this balance in different ways, we find that there is still some remnant site information present in the processed features by each of the MSDA techniques. Nevertheless, all MSDA methods produced latent features using which, it was difficult for the neural net to distinguish which site an instance was from. This ability to make features site-invariant is hypothesized to be the driving force behind improving the performance w.r.t. baseline performance.

## 5. Discussion and Conclusion

This paper analyses the performance of various existing MSDA models at classifying different objectives using rs-fMRI data, using data from popular public datasets ABIDE I and ADHD-200. We used FCMs of data as the representative feature vector upon which the models were trained and evaluated. Section 4 shows that the MSDA methods are successful in producing site-invariant latent features for the data, which in turn helps in improving classification accuracies. However, note that such methods are sensitive to the class distribution present in the data. While fixed hyperparameters provided useful insights into MSDA performances, [AR2:]an extensive hyperparameter tuning of the models was not included in this study owing to computational constraints. Furthermore, we found that some learning objectives were domain-invariant or unaffected by MSDA architectures (*e.g.*, age), [AR1:]however this was not exhaustively tested due to data limitations. Based on the experiments conducted, we observe that M³SDA consistently performed well across datasets and labels and was less prone to class imbalance. Models such as DARN, MDMN and MDAN performed better in the larger dataset (ABIDE I) and were sensitive to class imbalance, nevertheless, they performed significantly better when classes were balanced using simple sampling techniques. In general, we see that MDAN and M³SDA have improved the performance w.r.t. the baseline accuracies by a bigger margin than others for majority of the classifications. Based on these results, it is suggestive that MSDA techniques can be beneficial in improving the performance of DL techniques in neuroimaging-based applications.

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

## Appendix A. Domain Adaptation Methods

In the following sub-sections, a brief description of the underlying mechanisms in each of the MSDA techniques used in this study is given.

### A.0.1. Domain Adversarial Neural Networks (DANN)

Unlike rest of the methods used in this study, DANN (Ajakan et al., 2014) is a single-source domain adaptation technique that has been included to compare its performance with the other methods. DANN aims to learn features which help in accurately classifying data points based on their labels while making sure that the features are domain-agnostic. The architecture of the model used for adversarial training consists of three components, first, feature extractor $M_f$ which converts the input features to a latent representation (usually lower in dimension). The model bifurcates into two branches each of which is fed with these latent features. The first branch is the class predictor, $M_c$, which predicts the class to which the sample belongs to, whereas, the second branch is used for predicting in domain of the sample and is denoted by $M_d$. A gradient reversal layer (GRL), $G_\lambda(.)$, is attached at the start of $M_d$ to train the model in an adversarial fashion. The loss function which is minimized is given by:

$$E(\theta_f, \theta_d, \theta_c) = \sum_{s \in [1, S-1]} \sum_{i=1}^{N_s} \mathfrak{L}_c(M_c(M_f(x_s^i; \theta_f); \theta_c), y_s^i) +$$
$$\sum_{k \in [1, S]} \sum_{i=1}^{N_k} \mathfrak{L}_d(M_d(G_\lambda(M_f(x_k; \theta_f)); \theta_d), y_k^i) \tag{1}$$

$$\mathfrak{L}_c(\widehat{y}^i, y^i) = -y^i log(\widehat{y}^i) \tag{2}$$

$$\mathfrak{L}_d(\widehat{y}^i, y^i) = -(y^i log(\widehat{y}^i)) + (1 - y^i) log(1 - \widehat{y}^i)) \tag{3}$$

$$G_\lambda(\mathbf{X}) = \mathbf{X}, \ \frac{dG_\lambda}{d\mathbf{X}} = -\lambda \mathbf{I} \tag{4}$$

### A.0.2. Multisource Domain Adversarial Networks (MDAN)

MDAN can be seen as a logical extension to DANN wherein multiple source domains are used instead of one single source(Zhao et al., 2018). Apart from $M_f$ and $M_c$ the architecture contains one domain classifier $M_{d_i}$ for each of the $S-1$ source domains. The soft-max version of MDAN which utilizes the log-exp-max function to obtain a smoother approximation of the max function used in adversarial training was used in this work as it was shown to produce better and more computationally efficient results(Zhao et al., 2018). The model trains to minimize the following loss function:

$$E(\theta_f, \theta_{\mathbf{D}}, \theta_c) = \frac{1}{\gamma} log \sum_{s \in [1, S-1]} exp(\gamma(\mathfrak{L}_c^s + \mathfrak{L}_D^{s,S})) \tag{5}$$

$$\alpha_s = \frac{\mathfrak{L}_c^s}{\sum_{s \in [1, S-1]} \mathfrak{L}_c^s} \tag{6}$$

Where $\mathfrak{L}_c^s$ is the cross-entropy loss on label classification for the data from source site $s$ and $\mathfrak{L}_D^{s,S}$ is the domain discrimination loss for data from source site $s$ and target site $S$, similar to the losses defined in Eq.(1). The values of $\alpha_s$ are dynamically derived during the training process using each sites' losses helping the learning happen smoothly.

### A.0.3. DOMAIN AGGREGATION NETWORKS (DARN)

One of the main efforts in domain adaptation is during combining data from different sources, wherein, we need to select the domains which closely resemble the target domain in hand while excluding domains that are dissimilar. While inclusion of more domains provides the model with more data to train on, utilizing domains that are very different from the target domain leads to negative transfer (Jiménez-Guarneros and Gómez-Gil, 2020). DARN (Wen et al., 2020) aims at dynamically selecting and combining sites during the training phase to find the optimal selection in the trade-off between increasing sample size and decreasing negative transfer. DARN comprises of a feature extractor $M_f$, label classifier $M_c$, and a domain classifier $M_{d_i}$ for each domain. To define the objective function of the model, first, the model losses are defined by:

$$l_s(\theta_f, \theta_c, \theta_d) = \sum_{x^i, y^i \in D_s} \mathfrak{L}_c(M_c(M_f(x^i; \theta_f); \theta_c), y^i) + \\ \sum_{x^i, d^i \in D_s, D_S} \mathfrak{L}_d(M_{d_s}(M_f(x^i; \theta_d); \theta_c), d^i) \tag{7}$$

Consequently the collection of these losses for all sources is $\mathbf{L} = [l_1, l_2, \cdots, l_{S-1}]^\top$. A temperature parameter $\tau$ is incorporated with the final objective defined by:

$$\min_{\boldsymbol{\alpha} \in \triangle} - < \mathbf{z}, \boldsymbol{\alpha} > + \|\boldsymbol{\alpha}\|_2 \tag{8}$$

(define alpha's set) where $\mathbf{z} = \mathbf{L}/\tau$. To solve for the optimal $\boldsymbol{\alpha}$ values the Lagrangian dual of the above equation given by, $-\mathbf{z}^\top \boldsymbol{\alpha} + \|\boldsymbol{\alpha}\|_2 - \boldsymbol{\lambda}^\top \boldsymbol{\alpha} + \nu(\mathbf{1}^\top \boldsymbol{\alpha} - 1)$ for $\nu \in \mathbb{R}, \boldsymbol{\lambda} \geq 0$, is used. This gives us the optimal alpha values $\boldsymbol{\alpha}^*$ as:

$$\boldsymbol{\alpha}^* = \frac{[\mathbf{z} - \nu^* \mathbf{1}]_+}{\|[\mathbf{z} - \nu^* \mathbf{1}]_+\|_1} \tag{9}$$

where $\nu^*$ is found using binary search between $[min(\mathbf{z}) - 1, max(\mathbf{z})]$. Thus, the importance of each source domain is dynamic and keeps changing throughout the training phase to find the optimal sources and their contribution in developing the features for the target domain.

### A.0.4. Multi-Domain Matching Networks (MDMN)

MDMN also works on the concept of developing a shared feature space however the additional step is included to improve classification performance is based on mapping the feature space distributions of every source domain among themselves as well as mapping this common feature space to the target domain. The idea is to use a domain adapter which finds the degree of similarity between all the source domains so that the strength on similarity of the target domain is shared among all the similar source domains (Li et al., 2018c). The method allows all similar domains to merge together while keeping dissimilar domains away to reduce negative transfer. This is achieved by imposing a Wasserstein distance-based loss function which encourages the features from different domains to be closer to each other. MDMN has shown to help improve the classification performance while avoiding over-fitting issues as described in (Li et al., 2018c). The loss function which is used for training the model is given by:

$$E(\theta_f, \theta_d) = \frac{1}{SN_0} \sum_{i=1}^{N_0} \frac{1}{N_{s_i}} \mathbf{r}_{s_i}^T \mathbf{M}_d(M_f(x_i; \theta_f); \theta_d) \tag{10}$$

$$\mathbf{r}_s = \begin{cases} -\beta_s w_{ss'} & s' \neq s \\ \beta_s & s' = s \end{cases}, \forall s' \in [1, S] \tag{11}$$

Where, $N_{s_i}$ denotes the proportion of data that comes from $s_i$, and the data samples are taken in a mini-batch format given by $\{(\mathbf{x}_i, s_i)\} \forall i \in [1, N_0]$. In MDMN a single domain adapter is used with weight sharing instead of using $S$ different domain adapters for computational efficiency and is denoted by, $\mathbf{M}_d(; \theta_d) = [M_{d_1}(; \theta_d), M_{d_2}(; \theta_d), \cdots M_{d_S}(; \theta_d)]$. The definitions and strategies to calculate $\beta_s$ and $\mathbf{w}_s$ can be found in (Li et al., 2018c).

### A.0.5. Moment Matching for MSDA (M³SDA)

The main objective of M³SDA is to align the target domain with the source domains while aligning the source domains among themselves simultaneously during the training process. Unlike the other methods discussed, M³SDA tries to align the feature distribution moments of each source instead of using adversarial training for reducing the domain batch effects (Peng et al., 2019). One of the basic assumptions that this method is based on is that the posterior distribution of the class labels $P_{Y|X}$ would automatically align is the model is able to align the pior feature distributions $P_X$ of the domains. This assumption however might not hold true with practical datasets containing multiple sources. To mitigate this issue M³SDA-$\beta$ was introduced in (Peng et al., 2019), which has been used in this study as well. M³SDA-$\beta$ minimizes the domain discrepancy based on the $k^{th}$ order cross-moment divergence denoted by $d_{CM}^k(\cdot, \cdot)$, where k is a parameter taken as an input, furthermore the training strategy utilized in (Peng et al., 2019) was applied for this model. The main loss function can be written as:

$$E(\theta_f, \theta_c) = \sum_{s \in [1, S]} \sum_{i=1}^{N_s} \mathfrak{L}_d(M_c(M_f(x_s^i; \theta_f); \theta_c), y_s^i) + d_{CM}^k(F_s, F_S) \tag{12}$$

where $F_s, F_S$ denote the feature vectors received from $M_f$ for the source and target data $\mathbf{x}_s$ and $\mathbf{x}_S$ respectively. Apart from the feature extractor $M_f$, M³SDA-$\beta$ also uses a pair

of classifiers $M_C$ and $M_{C'}$ for each domain denoted by $\mathbf{M}_{C'} = [(M_{C_1}, M_{C'_1}), (M_{C_2}, M_{C'_2}), \cdots, (M_{C_S}, M_{C'_S})]$. The training strategy involves using $M_f$ and $\mathbf{M}_C$ in a three-step process wherein, first, both the models are trained together to classify multi-source samples. Next, $\mathbf{M}_C$ is trained while keeping $M_f$ fixed to maximize the target domain discrepancy between each of the classifiers in a classifier pair from $\mathbf{M}_C$. Finally, $M_f$ is trained while fixing $\mathbf{M}_C$ to minimize the discrepancy of each classifier pair in $\mathbf{M}_C$. The process repeats until convergence is achieved and during testing a weighted-average of the classifier outputs is used to make predictions, wherein the weights are defined using source-only accuracies as described in (Peng et al., 2019).

## Appendix B. Data Demographics

The ABIDE I dataset (Craddock et al., 2013a) is a combination of fMRI scans from 17 different sites. The dataset provides users with rs-fMRI, T1 structural brain images and phenotypic information for each patient. It consists of 505 ASD scans and 530 controls. As a part of pre-processing the rs-fMRI data, the C-PAC processing pipeline offered by Preprocessed Connectome Project (Craddock et al., 2013b) was used. The pipeline consists of several steps such as slice-time correction, motion correction, intensity normalization, and nuisance signal removal. Furthermore, data from all the sites were spatially registered to the MNI152 template space, along with being passed through a band-pass filter (0.01 - 0.1Hz) to remove any high frequency noise in the data The site-wise distribution of age and sex is described in $^{AR2:}$~~Table B~~ Table 1.

Similarly, the ADHD-200 dataset(Bellec et al., 2017), which was first introduced during the ADHD-200 Competition, contains scans from 8 different sites which inclues a total of 973 individuals. This dataset also provides one or more rs-fMRI, T1 structural MRI and the respective phenotype for each individual.The scans undergo similar preprocessing using a pipeline made available by Neuroimaging Analysis Kit (NIAK) which includes steps such as Slice timing correction, motion correction, linear and non-linear spatial normalization, correction of physiological noise, Spatial smoothing and MNI T1 space registration. . The distribution of the data according to the phenotype is provided in $^{AR2:}$~~Table B~~ Table 2. Since, the phenotypic data had inconsistent and missing age information, the particular column has been omitted from the table.

## Appendix C. Model Specifications

The architecture of the various components in each of the pipelines was kept constant, i.e. the feature extractor, label classifier and domain adapter had the same design across all methods. Since the features were flattened FCM, fully connected layers (FCN) were used along with dropouts and L-2 regularization. The sub-models' designs are described in Table 3. $^{AR4:}$In most cases, the complete model is trained end-to-end using using the Adam optimizer (Kingma and Ba, 2014) on the loss functions defined in Appendix A. Few of the models (*e.g.*, MDMN) utilize a training strategy unlike other methods (see Appendix A). In such methods, the training process described by in the original papers are utilized. The hyperparameters used are listed in Table 4 and Table 5. Common hyperparameters

| | ASD | | TC | |
|---|---|---|---|---|
| Sites | Age | Sex | Age | Sex |
| Pitt | 19.0 (7.3) | M 25, F 4 | 18.9 (6.6) | M 23, F 4 |
| Olin | 16.5 (3.4) | M 16, F 3 | 16.7 (3.6) | M 13, F 2 |
| OHSU | 11.4 (2.2) | M 12, F 0 | 10.1 (1.1) | M 14, F 0 |
| SDSU | 14.7 (1.8) | M 13, F 1 | 14.2 (1.9) | M 16, F 6 |
| Trinity | 16.8 (3.2) | M 22, F 0 | 17.1 (3.8) | M 25, F 0 |
| UM | 13.2 (2.4) | M 57, F 9 | 14.8 (3.6) | M 56, F 18 |
| USM | 23.5 (8.3) | M 46, F 0 | 21.3 (8.4) | M 25, F 0 |
| Yale | 12.7 (3.0) | M 20, F 8 | 12.7 (2.8) | M 20, F 8 |
| CMU | 26.4 (5.8) | M 11, F 3 | 26.8 (5.7) | M 10, F 3 |
| Leuven | 17.8 (5.0) | M 26, F 3 | 18.2 (5.1) | M 29, F 5 |
| KKI | 10.0 (1.4) | M 16, F 4 | 10.0 (1.2) | M 20, F 8 |
| NYU | 14.7 (7.1) | M 65, F 10 | 15.7 (6.2) | M 74, F 26 |
| Stanford | 10.0 (1.6) | M 15, F 4 | 10.0 (1.6) | M 16, F 4 |
| UCLA | 13.0 (2.5) | M 48, F 6 | 13.0 (1.9) | M 38, F 6 |
| Maxmun | 26.1 (14.9) | M 21, F 3 | 24.6 (8.8) | M 27, F 1 |
| Caltech | 27.4 (10.3) | M 15, F 4 | 28.0 (10.9) | M 14, F 4 |
| SBL | 35.0 (10.4) | M 15, F 0 | 33.7 (6.6) | M 15, F 0 |

Table 1: ABIDE I demographics. The age is represented by the mean (standard deviation) format and the sex distribution is denoted by M: males and F: females.

| | ADHD | TC |
|---|---|---|
| Site | Sex | Sex |
| KKI | M 15 F 10 | M 41 F 28 |
| *AR2:NI*NeuroImage | M 31 F 5 | M 12 F 25 |
| NYU | M 117 F 34 | M 56 F 55 |
| OHSU | M 30 F 13 | M 30 F 40 |
| Peking | M 92 F 10 | M 84 F 59 |
| Pittsburg | M 3 F 1 | M 50 F 44 |
| UWash | M 0 F 0 | M 33 F 28 |
| Brown | M 0 F 0 | M 9 F 17 |

Table 2: ADHD-200 demographics

(shown in Table 4) are kept constant throughout all experiments and datasets, while specific hyperparameters (see Table 5) are selected from a range of candidate values.

## Appendix D. Class Balancing

As discussed in Section 4, to handle the data imbalance the minority class was over-sampled to match in number with the majority class. In case a particular site consists of data of only

| Sub-Model | Architecture | Output |
|---|---|---|
| Feature Extractor ($M_f$) | input $\rightarrow$ 2000 $\rightarrow$ 1000 | latent features |
| Label Classifier ($M_c$) | 1000 $\rightarrow$ 100 $\rightarrow$ 2 | label predictions |
| Domain Adapter ($M_d$) | 1000 $\rightarrow$ 100 $\rightarrow$ n* | class predictions |

Table 3: FCN architectures used in each of the sub-models. Each layer was followed by a ReLU layer (except last layer where softmax is used) and a dropout layer with p=0.5. The output of $M_d$ has different number of nodes for different models and datasets and therefore, is represented by "n".

|  | ABIDE I | ADHD-200 |
|---|---|---|
| $\alpha$ | 0.0001 | 0.0003 |
| batch_size | 100 | 200 |
| dropout | 0.5 | 0.5 |
| epochs | 100 | 50 |

Table 4: List of hyperparameters $^{AR2:}$which were kept constant throughout all the experiments and used in all MSDA methods. $^{AR2:}$'*' denotes that that hyperparameter may be different for different models based on the hyperparameter tuning conducted, hence, the value that was used mostly has been reported.

| Dataset | Hyperparameters | Illness | | | | | Sex | | | | | Age | | | | |
|---|---|---|---|---|---|---|---|---|---|---|---|---|---|---|---|---|
|  |  | DANN | DARN | MDAN | MDMN | M³SDA | DANN | DARN | MDAN | MDMN | M³SDA | DANN | DARN | MDAN | MDMN | M³SDA |
| ABIDE I | $\mu$ | 0.1 | 0.01 | 0.1 | 0.1 | 0.01 | 0.1 | 0.1 | 0.1 | 0.01 | 0.1 | 0.1 | 0.1 | 0.1 | 0.01 | 0.1 |
|  | $\gamma$ | - | 0.7 | 5 | - | - | - | 0.5 | 5 | - | - | - | 0.7 | 10 | - | - |
| ADHD-200 | $\mu$ | 1 | 0.001 | 0.01 | 1 | 1 | 0.1 | 1 | 1 | 0.1 | 1 | - | - | - | - | - |
|  | $\gamma$ | - | 0.5 | 2 | - | - | - | 0.5 | 3 | - | - | - | - | - | - | - |

Table 5: $^{AR2:}$model-specific hyperparameters used in our experiments apart from the common hyperparameters in Table 4.

one class, the site is dropped in that experiment (*e.g.*, ADHD illness classification displays 6 sites instead of 8). To understand the impact of class balancing on improving performance of each model, the comparison of the accuracies before and after data balancing is provided in Figure 5. The data in ABIDE I for illness already contained balanced classes and hence was omitted.

It is seen in Figure 5 balancing data in the case of age classification for ABIDE Imade no difference when it came to MSDA performances, while we see a significant improvement while using this strategy for sex classification with an increase of as high as 8% for the MDMN model. The accuracy changes in ADHD-200 are quite different than what is observed for the ABIDE dataset. We see that in the case of illness classification, all models (except DANN) benefited from the balancing. An interesting observation can be made in sex classification for ADHD-200, wherein the accuracy scores of MDAN increased by almost

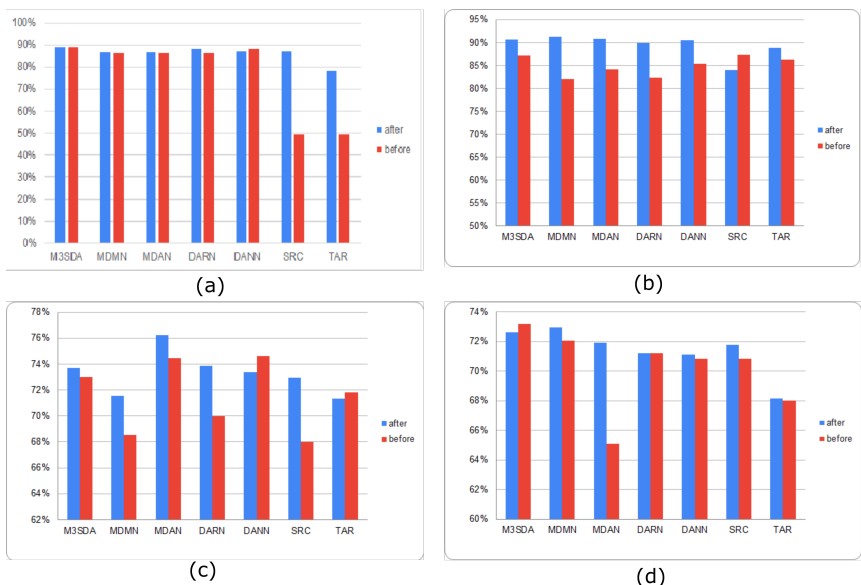

Figure 5: A comparison between the accuracies before and after balancing the classes using oversampling. The age and sex classifications for ABIDE I are shown in (a) and (b), while (c) and (d) represent the illness and sex classifications for ADHD-200 dataset.

7%. Hence, in most case data balancing had a postive and significant impact at improving model performance.

