# OpenReview forum: "Multi-source Domain Adaptation Techniques for Mitigating  Batch Effects: A Comparative Study"
_MIDL.io/2021/Conference — Submitted to MIDL 2021_

### Official Review · AnonReviewer3 · 2021-03-01

**Confidence:** 4
**Preliminary Rating:** 3
**Final Rating:** 2

**Summary:**

This paper considers an important batch-effect or domain shift problem in fMRI-based brain disease diagnosis and introduces the empirical evaluation of various domain adaption methods in the literature. From their experiments, it was reported that the M3SDA generally performed the best and an MSDA approach in general helps improve accuracy.

**Strengths:**

The data harmonization problem in fMRI-based disease diagnosis is of great importance in the field.
The empirical comparison of various domain adaptation methods over two datasets provides meaningful insights and results.

**Weaknesses:**

In order for better use and reproducibility of the results, it would be great to open their codes to the public.
A more rigorous analysis and interpretation of the results is highly recommended to better justify their results and conclusions.

**Deanonymize Review:**

no

**Detailed Comments:**

In order for better legibility, it is recommended to describe Fig. 2 in detail. For example, what does the horizontal black line mean? In caption, are the dataset names of (a) and (b) misaligned?

Page vii, Age-prediction task: "The baseline and MSDA models both have an accuracy of around 90-91%." -> 87%(?)

In regard to Fig. 4 and the related explanation in the main body, to this reviewer’s interpretation, the trained feature representation seems still have some site-related information because the classification accuracy is still much higher than the chance level in each site, i.e., 1/17 (ABIDE) and 1/8 (ADHD).

**Final Rating Justification:**

As a validation/application paper, it was recommended to conduct more rigorous analysis and interpretation of the results because such additional information could be the contribution of the work. However, their strategy of adding plots and insights in the Appendix is not suitable to keep my initial rating. In this regard, I would like to decrease my rating.

**Justification Of The Preliminary Rating:**

This paper introduces an important milestone for the multi-site data-harmonization problem in fMRI-based disease diagnosis and cognitive identification. The empirical comparison of various domain adaptation methods over two datasets provides meaningful insights and results.

**Paper Type:**

validation/application paper

**Questions To Address In The Rebuttal:**

Please refer to the Detailed Comments section.

**Special Issue:**

no

---

> ### Author Response · Authors · 2021-03-18
> **Response to AnonReviewer3**
>
> > 1) It would be great to open their codes to the public. A more rigorous analysis and interpretation of the results is highly recommended to better justify their results and conclusions.
>
> * We agree with your suggestion and we plan to release the codebase and data splits soon after cleaning the code and adding the necessary information required to reproduce results. We will also add a few more plots and insights in the Appendix for more clarity. Thank you for your recommendation.
>
> > 2) ... For example, what does the horizontal black line mean? In caption, are the dataset names of (a) and (b) misaligned?
>
> * Thank you for bringing this to our attention. The names in Fig. 2 were flipped by mistake and they have been fixed in this revision. The black line represents the SRC model’s accuracy score and is referred to as target-line in the caption of Fig. 2, we shall definitely make the description legible and clearer by the next revision.
>
> > 3) ... the trained feature representation seems to still have some site-related information because the classification accuracy is still much higher than the chance level in each site, i.e., 1/17 (ABIDE) and 1/8 (ADHD).
>
> * The accuracies in Fig. 4 represent the values obtained after conducting a 10 fold cross-validation on data combined from all sites with "Site" as the label and taking the average of the accuracy of all folds. While it can be said that there is still some site information left after using the MSDA methods, we can note that the accuracies in site classification are lower than the baseline to indicate that MSDA models do harmonize the features extracted from fMRI data to some extent. We believe the reason that there is remnant site information present even after domain adaptation is because MSDA models try to make the features domain-invariant while making sure they retain their discriminatory properties w.r.t. the labels they are meant to be classified upon. Hence, owing to this tradeoff, the model is not able to completely eradicate all of the site information.  The normalized confusion matrix or a graph with site-wise accuracies for these 10-folds will be added to the revised paper for better insights on the site-classification performance.  Furthermore, we have added a few lines discussing this observation in Section (4).
>
> We would like to thank the reviewer for the time they spent and their insightful comments, it has helped us in making our paper more refined and detailed.

---

### Official Review · AnonReviewer1 · 2021-03-08

**Confidence:** 4
**Preliminary Rating:** 3
**Recommendation:** Oral
**Final Rating:** 3

**Summary:**

This paper empirically analyses the performance of different MSDA methods (MDAN, DARN, MDMN, M3SDA) for classification tasks using rs-fMRI data that are from public datasets ABIDE I and ADHD-200. It also evaluated the impact of class imbalance and number of sites. The results showed that MSDA models presented an effectiveness in neuroimaging-based applications.

**Strengths:**

1. This paper conducted abundant experiments to prove the effectiveness of MSDA methods in deal with classifying different objectives using rs-fMRI data.
2.  The paper describes the principle of MSDA methods in details, which is easy to follow.
3.  It is a good idea that the paper reports the results of a two-layered fully-connected network that was trained and tested to classify the sites based on input latent features to imply that the latent features have become site-invariant after using MSDA.


**Weaknesses:**

1.  The description of ABIDE I and ADHD-200 was confusing. In section 3.1, the authors said that ABIDE I has 17 different sites and ADHD-200 has 8 sites. However, in Fig. 2, results of 6 sites were shown for  ABIDE I and 17 sites for ADHD-200.
2. Details of classification of input latent features should be given. How to collect the training data? What was the split of datasets for training and testing?


**Deanonymize Review:**

no

**Final Rating Justification:**

The authors' reply has clarified some points. However, the implementation of the compared methods seems to be suboptimal, due to using default hyper-parameters. I would keep my original rating.

**Justification Of The Preliminary Rating:**

This is a validation paper and the experiments with various domain adaptation methods and two different datasets are extensive. They authors described what they had done clearly. Therefore, I suggest weak accept.

**Paper Type:**

validation/application paper

**Questions To Address In The Rebuttal:**

1.  In the conclusion, “Furthermore, we found that some learning objectives were domain-invariant or unaffected by MSDA architectures (e.g., age).” This claim should also be validated on the SDHD-200 dataset.

2. As age and sex both belong to patient demographics. The MSDA methods had some effect for the sex classification but not effective for age classification.  Some discussions and explanations are expected.

**Special Issue:**

no

---

> ### Author Response · Authors · 2021-03-18
> **Response to AnonReviewer1**
>
> > 1)  In section 3.1, the authors said that ABIDE I has 17 different sites and ADHD-200 has 8 sites. However, in Fig. 2, results of 6 sites were shown for ABIDE I and 17 sites for ADHD-200.
> * Sorry, the labels were flipped by mistake; this will be fixed in the next draft. Thank you for pointing this out. The reason that only 6 sites are shown in ADHD-200 illness classification is because, as a part of our class balancing step, the sites which contain data with one single class are dropped during that classification. Sites BROWN and UWash both contain only healthy control data points, which would hamper the MSDA techniques performance and hence, were dropped for illness classification.
>
>
> > 2)  How to collect the training data? What was the split of datasets for training and testing?
> * The data was collected using python packages, mainly `Nilearn` and the splits were based on a manual seed set in the code. The official code repo will be released soon after the final revision, wherein we can share the splits, data acquisition and model training-testing codes used in this work.
>
> > 3)  “Furthermore, we found that some learning objectives were domain-invariant or unaffected by MSDA architectures (e.g., age).” This claim should also be validated on the SDHD-200 dataset.
> * Thank you for your kind suggestion. The main reason we were not able to perform age classification on ADHD-200 was owing to the unavailability of the ‘age’ phenotype for this dataset. As we did not have access to the labels, we had to omit the particular classification from this study (mentioned in Appendix B).
>
> > 4) The MSDA methods had some effect for the sex classification but not effective for age classification. Some discussions and explanations are expected.
> * We agree with your review on MSDA’s performance for patient demographics. One particular observation that could be noted is that in the case of ABIDE dataset is that MSDA was not able to significantly improve the accuracy from baseline for both sex and age classification. In the case of ABIDE, we see that except for M3SDA, the MSDA models performed, as well as or worse, than baseline performance. However, for ADHD-200, MSDA models were able to significantly improve the accuracy over the baseline. This may suggest that the observation could be limited to just the ABIDE dataset and might differ for other datasets accordingly. Since an exhaustive experimentation on this aspect would not be feasible for this paper, the revised paper will  acknowledge it as a limitation of the study (see Section 5).
>
> Thank you once again for helping us better our work, we are grateful for your contributions.

---

### Official Review · AnonReviewer4 · 2021-03-08

**Confidence:** 4
**Preliminary Rating:** 2
**Final Rating:** 2

**Summary:**

This paper presents a study to compare unsupervised multi-source domain adaptation (MSDA) strategies for handling multi-site effects from resting-state fMRI data gathered from multiple imaging sites. Five different MSDA methods were tested and compared against no adaptation of the source data and a model trained on only target data, using 2 public resting-state fMRI datasets, ABIDE I and ADHD-200, and 3 different prediction tasks. The work found that MSDA methods generally performed better, although with potential sensitivity to class imbalance or dataset, and it was concluded that the M3SDA technique (moment-matching for SDA) had the most consistently good performance across datasets and tasks.

**Strengths:**

1. This validation paper uses 2 large public resting-state fMRI datasets for benchmarking and compares many different unsupervised MSDA approaches, using 3 different classification tasks.

2. Statistical significance testing was performed between tested methods and baseline no adaptation model to assess differences in model performance.

3. The diagram and plots are effective in displaying the information for easy digestion.

4. The paper is well organized and easy to follow.

**Weaknesses:**

1. The training and testing splits are well explained, but how exactly the models are trained is not well described. Are there important hyperparameter settings for any of the MSDA methods that needed to be tuned? Is a validation dataset used to tune model parameters? How about optimization settings?

2. I have some concerns about the baseline, no adaptation source approach for ABIDE dataset, in which Fig. 3 shows an overall accuracy of >80% for classification of autism vs. healthy. I have never before seen an accuracy this high under leave-one-site-out cross-validation before (generally accuracies hover at 70% or below). I realize that for each site only 1 out of 10 folds of the target site data is being used for testing, but this jump in accuracy is rather surprising, and thus triggers some skepticism for the soundness of the overall experiments.

3. The authors discuss how creating more site invariant features should result in improved out-of-domain performance, with Fig. 4 as evidence of learning more site invariant features than the original source-only model. However, the model which shows the lowest site classification accuracy and thus is most successful in reducing site-dependent features (DARN) actually appears to perform the worst overall in the various classification tasks, as seen in Fig. 3.  Furthermore, the method that actually made the features more site-identifiable (DANN) appears to perform better than DARN (we can see 2 cases where DANN produced significant differences in classification accuracy whereas DARN did not). Thus, I am wondering whether there is a problem somewhere in the trained model for the classification tasks, or a problem with the model for classifying the site invariant features, or whether the conclusion about site-invariant features improving performance is not correct (although arguably this is what is driving much of the domain adaptation problem). It does seem strange that post MSDA, the domain classifier still has such high accuracy.

4. While the paper flow is good, there are many typos throughout that are distracting. A solid proofreading of the paper should fix this problem, though.

**Deanonymize Review:**

no

**Detailed Comments:**

1. Important typo: in Fig. 2, the captions for (a) and (b) appear to be switched ((a) should be for ADHD, (b) for ABIDE).

2. It would be helpful if the caption for Fig. 2 specifically noted that the results were for illness classification.

3. I believe the AAL atlas should be 116 labels (this is what is listed for ABIDE I preprocessed data). Is the 119 regions in the paper a typo?


**Final Rating Justification:**

While I appreciate the authors' responses and some of the helpful clarifications, there are still some missing information regarding the tuning of hyperparameters (as also noted by another review), and it seems likely that these hyperparameters were tuned based on test set since there is no mention of validation set in the paper, which I would say is unacceptable for a validation paper (and may explain some of the unusually high accuracies) as this wouldn't accurately reflect the performance of each model. Also the point on the site classification - I can accept that the method that does better site harmonization may be sacrificing target task results, but it seems strange and unexplained that a method aiming to harmonize the data leads to higher site classification than the src only model. Therefore I keep my original rating.

**Justification Of The Preliminary Rating:**

The paper provides an interesting study with 2 public datasets comparing unsupervised domain adaptation techniques for handling multi-site data, and also provides a nice overview of some existing MSDA methods. However, some concerns about the results and lack of some details leads to some reduced confidence in the experimental results section.

**Paper Type:**

validation/application paper

**Questions To Address In The Rebuttal:**

In particular, if the authors could respond in particular to points 2 and 3 in the "Weaknesses" section, I would appreciate it.

**Special Issue:**

no

---

> ### Author Response · Authors · 2021-03-18
> **Response to AnonReviewer4**
>
> > 1) Are there important hyperparameter settings for any of the MSDA methods that needed to be tuned? Is a validation dataset used to tune model parameters? How about optimization settings?
>
> * In general, all of the MSDA techniques are trained end-to-end, based on the loss functions described in Appendix (A), except for a few methods where a different process needed to be applied (e.g.,MDMN). Common hyperparameters (e.g., #epochs, batch size, $\alpha$) are set to the standard values without extensive fine-tuning. Model-specific hyperparameters such as $\mu$ and $\gamma$ are selected from a range of candidate values suggested by the developers of the techniques. Please see our replies for AnonReviewer2 about the hyperparameters for further information. Adam optimizer has been used for parameter-optimization (info. added in Appendix C.) In general we noted that the models were robust, and performed well for a wide range of hyper-parameters
>
> > 2) I have some concerns about the baseline, no adaptation source approach for ABIDE dataset, in which Fig. 3 shows an overall accuracy of >80% for classification of autism vs. healthy. I have never before seen an accuracy this high ...
>
> * This is a good observation. The other analyses all pooled the 17 different sites, to produce a micro-average (in essence, averaging the scores for each site, proportional to the size of that site). By contrast, our analysis provides a macro-average: just taking the (raw) average over the 17 sites.  As our accuracy results are higher for the smaller sites, our overall macro-average score is higher than the others’ micro-average. Furthermore, as the reviewer has rightly mentioned, the plotted accuracies are an average of the fold accuracies for each site and since each fold has a lesser number of test points, the final score is higher.
>
> > 3) …  I am wondering whether there is a problem somewhere in the trained model for the classification tasks, or a problem with the model for classifying the site invariant features, or whether the conclusion about site-invariant features improving performance is not correct …
>
> * Thank you for your justified, and fascinating, observation. Our experiments show there is a tradeoff between classification accuracy and domain invariance. Sometimes, making the features domain-invariant came at the cost of reduced classification accuracy. An ideal MSDA algorithm would either strike a balance between these two properties by itself or it would retain or increase the discriminatory features of the data while harmonizing the sites. Since each method uses its own strategy to do this, some models perform better while making the features domain invariant while retaining information useful for classification, as compared to others. That can explain why DARN’s site harmonization was the highest but sometimes scored worse than DANN and other models in classification tasks. We will incorporate this interesting insight into the paper. Thank you for pointing this out to us.
>
> > 4) While the paper flow is good, there are many typos throughout that are distracting. A solid proofreading of the paper should fix this problem, though.
>
> * We thank you for bringing this to our notice. This revision will fix the typos present in the paper.
>
>
> Thank you very much for providing us with your observations and recommendations. We were able to add new insights and improve the paper's quality, we appreciate your contribution.

---

### Official Review · AnonReviewer2 · 2021-03-09

**Confidence:** 3
**Preliminary Rating:** 1
**Final Rating:** 1

**Summary:**

This paper compares various techniques for multi-source domain adaptation which are designed to mitigate batch effects in rs-fMRI datasets with heterogenous sources. Specifically, the authors compare four techniques, namely MDAN, DARN, MDMN, and M3SDA for classification of disorder, age, sex on two different real world public datasets, i.e. ABIDE I and ADHD

**Strengths:**

As a validation/application paper, the topic - multi source domain adaptation is potentially of broad interest to a wider audience, even beyond rs-fMRI analysis. Mitigating batch effects is an important area of study, and having a comprehensive comparison of various existing methods addressing this problem could provide useful for future work. The paper is also fairly clear in its presentation of methods.

**Weaknesses:**

The main weakness of the paper is the experimental evaluations, as many details are either missing, ambiguous or not adequately explained. The authors mention they report average 10 fold cross validation performance without a validation set to determine hyperparameters. Moreover, Table 4 suggests that different parameters were used on individual datasets. It sounds like best 10 fold CV performance was reported for each dataset, i.e. the results are still biased since the same data was used to generate hyperparameters. There are also inconsistencies with the reporting of site-specific results within the paper.

**Deanonymize Review:**

no

**Detailed Comments:**

Clarifications:

1. How exactly are the hyperparameters tuned for each method, both those reported in Table 1 with an asterisk and without the askterisk?

2. What result is being displayed in Fig. 2 i.e. which classification task(s)? Why are the ABIDE and ADHD-200 site labels different from Table 1 and 2? What does the "Neuroimage" site correspond to in Fig. 1 a?

2. How are the age prediction labels determined? Table 1 suggests that several ABIDE sides have a small and narrow age range with the study mainly focus on adolescents - for example KKI. What does an age classification of young vs old mean in this setting? How does it perform? Does this contribute to the lack of improvements provided by the MSDA models?

3. How does each model compare in terms of learnable hyperparameters?

4.  Appendix B references Table B, but there isn't a Table B in the paper or Appendix.


**Final Rating Justification:**

I thank the authors for adding some clarifications. However, they have not sufficiently clarified a few points, the first being the the hyperparameter selection. Also, the authors response to my comment about site-specific metrics to strengthen the argument of the paper has been deferred to the post-acceptance stage. In my opinion, this is not a viable strategy. Given that this is a validation/application paper, I would like to keep my original rating.

**Justification Of The Preliminary Rating:**

Considering that this is a validation/application paper, there are far too many unexplained points, and inconsistencies for its acceptance as a full paper in the current form. The experimental results seem incomplete and the supporting discussion not convincing enough. It also sounds like the authors did not use best practices for hyperparameter selection of the models, which makes me question the validity of the comparisons being made.

**Paper Type:**

validation/application paper

**Questions To Address In The Rebuttal:**

1. The authors make a case for DA by comparing against the SRC model. However, they do not compare against any baselines that aren't a part of the methods examined in the paper. Could the authors please put the result (averaged and site specific) in the context of state-of-the-art classifiers on ABIDE I and ADHD-200 for (at least) disease classification?

2. Along with this, I would also like to know how individual hyperparameters for each models were tuned.

4. Fig. 4 reports only the accuracy metric for determining whether the features are domain-invariant. In general, given that different sites have different number of examples and the setting is multi-class classification, I'd be curious to know how either site-specific accuracies, and/or sensitivity vs specificity of this classification task looks like. Additionally, what classification algorithm was used here?

**Special Issue:**

no

---

> ### Author Response · Authors · 2021-03-18
> **Response to AnonReviewer2 [1/2]**
>
> Thank you very much for the detailed and insightful feedback. Your review has allowed us to greatly improve the paper's quality and include further useful information.
>
> > 1) How exactly are the hyperparameters tuned for each method, both those reported in Table 1 with an asterisk and without the asterisk?
>
> * Since the intention of this study was to compare the performance of the MSDA techniques in their default settings, extensive hyperparameter tuning was avoided.
>
> * For most of the experiments, we just used the frequently-used values of the hyperparameters –  eg, [1], [2], [3] all use the same $\alpha$ or batch sizes.
> * We did need to change a few of the hyper-parameters, such as $\alpha$ and epochs for ADHD, to expedite the learning process and reduce the training-time; we found this had only a minimal impact on accuracy scores. The main hyper-parameters that needed to be changed for specific models were $\mu$ and $\gamma$, as different models were sensitive to different ranges of values for these hyper-parameters. The values were chosen from a range of candidate settings that were based around the values generally used in these techniques in previous literature.
> * While one might be able to improve the performance by adjusting the hyperparameters further, as mentioned above, we decided to use the standard values, as it was a computationally expensive option to tune all the hyperparameters and also because the data in target site needed to be used for both training (9 folds) and testing (1 fold), removing 1 fold for validation would reduce the amount of data available for training and hamper its performance. Furthermore, the revised paper includes Table 5, which provides the hyperparameters that were changed for each MSDA method. Note that:
>   * Only DARN and MDAN utilized $\gamma$, which was used in the range as suggested by the developers of these methods.
>   * $\mu$ denotes the fraction of domain adaptation loss in the total loss. This hyper-parameter was also selected for a set of candidate values ( mostly {1, 0.1, 0.01}) and we found that the accuracies did not change much when for wide ranges of $\mu$ around chosen values.
>
> > 2)  What result is being displayed in Fig. 2 i.e. which classification task(s)? Why are the ABIDE and ADHD-200 site labels different from Table 1 and 2? What does the "Neuroimage" site correspond to in Fig. 1 a?
>
> * We thank the reviewer for bringing this error to our notice. Yes, the names were jumbled and the labels were inverted; these are fixed in the revised version. Similarly, the revised version will match the order of the names in the figure and the table. It will also note that “Neuroimage” (one of the sites present in the ADHD-200 dataset) corresponds to “NI” in the table.
>
> > 3) How are the age prediction labels determined? Table 1 suggests that several ABIDE sides have a small and narrow age range with the study mainly focused on adolescents - for example KKI. What does an age classification of young vs old mean in this setting? How does it perform? Does this contribute to the lack of improvements provided by the MSDA models?
>
> * As mentioned in Section 3.1, we defined Age=Old (resp., Young) if the patient’s age is above (resp., below) the median age, with respect to the complete dataset. As you correctly noted, this does make a few of the sites imbalanced (~ 5 out of 17). However, based on the number of samples of each class, the data is balanced using the technique described in Appendix D. Furthermore, the 10-folds are made in a stratified manner, to make sure that the fold represents each of the classes in a manner appropriate  for the testing domain. The revised version will describe this.
>
> ## References
> [1] Nigri, Eduardo, et al. "Explainable deep CNNs for MRI-based diagnosis of alzheimer’s disease." 2020 International Joint Conference on Neural Networks (IJCNN). IEEE, 2020.
>
> [2]  Hon, Marcia, and Naimul Mefraz Khan. "Towards Alzheimer's disease classification through transfer learning." 2017 IEEE International conference on bioinformatics and biomedicine (BIBM). IEEE, 2017.
>
> [3]  El Gazzar, Ahmed, et al. "Simple 1-D convolutional networks for resting-state fMRI based classification in autism." 2019 International Joint Conference on Neural Networks (IJCNN). IEEE, 2019.

---

> ### Author Response · Authors · 2021-03-18
> **Response to AnonReviewer2 [2/2]**
>
> > 4) Appendix B references Table B, but there isn't a Table B in the paper or Appendix
>
> * We apologize for this confusion. This error is fixed in the revision.
>
> > 5) The authors make a case for DA by comparing against the SRC model. However, they do not compare against any baselines …
>
> * While comparison with SOTA is valuable, the training and testing paradigms used in this paper are fundamentally different from those generally used for such studies. Previous SOTA learners reported their scores in either a leave-one-site-out strategy or by mixing the data points from all sites in the dataset and splitting it into train-test. However, this cannot be done in our framework as we require (1) using training data from different sites (while retaining site information) and (2) unlabeled data from the target domain. Hence, a direct comparison of the accuracies might not be feasible.
>
> > 6) I'd be curious to know how either site-specific accuracies, and/or sensitivity vs specificity of this classification task looks like. Additionally, what classification algorithm was used here?
>
> * The main text provides accuracies; by the final revision, the appendix will include a normalized confusion matrix or a graph with site-wise accuracies for the classifications, to show the distribution of predicted labels for each of the models. The classification algorithm is described in Section 4, which uses a 2-layered fully connected neural network for classification.

---

### Meta-Review · Area_Chair1 · 2021-03-28

**Recommendation:** Reject

**Metareview:**

As noted by most reviewers, as a validation/application paper, a specific emphasis should be given to provide a rigorous experimental design. However this crucial part of the paper appears quite unclear and the possibly dubious and not clarified way the hyperparameters are tuned make the current results difficult to accept at face value.

**Paper Type:**

validation/application paper

---

### Decision · Program_Chairs · 2021-03-31

Reject